

# On the parametrization of optical particle counter response including instrument-induced broadening of size spectra and a self-consistent evaluation of calibration measurements

Adrian Walser[1,2], Daniel Sauer[2], Antonio Spanu[3], Josef Gasteiger[3], and Bernadett Weinzierl[3]

[1]Ludwig Maximilians University (LMU), Meteorological Institute, Munich, Germany
[2]German Aerospace Center (DLR), Institute of Atmospheric Physics, Oberpfaffenhofen, Germany
[3]University of Vienna (UNIVIE), Aerosol Physics and Environmental Physics, Vienna, Austria

*Correspondence to:* B. Weinzierl (bernadett.weinzierl@univie.ac.at)

**Abstract.** Optical particle counters (OPCs) are common tools for the in situ measurement of aerosol particle number size distributions. As the actual quantity measured by OPCs is the intensity of light scattered by individual particles, it is necessary to translate the collected distribution of scattering signals into the desired distribution of particle sizes. A crucial part in this challenge is the modeling of OPC response and the calibration of the instrument, i.e. to establish the relation between particle

5    scattering cross-section and measured signal amplitude. To date, existing methods lack a comprehensive parametrization of this instrument response, particularly regarding the instrument-induced broadening of size distribution widths. We introduce an approach overcoming the present shortcomings by implementing a simple parametrization of the broadening effect and a self-consistent way to evaluate calibration measurements using a Markov chain Monte Carlo (MCMC) method. We further outline how to obtain realistic uncertainty estimates for OPC size distributions within this new framework. Measurements of

10   particle standards for two OPCs, the Grimm model 1.129 (SkyOPC) and the DMT Passive Cavity Aerosol Spectrometer Probe (PCASP), show substantial reduction in residuals between measured and modeled response compared to existing methods. For the presented set of measurements only the new approach yields results that are conform with the true size distributions within the range of model uncertainty. The offered approach will help to improve the accuracy of derived size distributions and the assessment of their precision for OPC measurements in general.

## 15  1  Introduction

The size distribution of aerosol particles is a key property to understanding the impact of aerosols on human health and Earth's climate. To measure aerosol size distributions optical particle counters (OPCs) are widely used in air quality programs and atmospheric studies. However, several studies directly comparing size distributions from different OPC instruments (e.g. Belosi et al., 2013; Renard et al., 2016) and OPCs with other sizing methods (e.g. Reid et al., 2003; Müller et al., 2012) find significant

20  disagreements and in some cases OPCs show systematic mis-sizing and artificial broadening of size spectra. This highlights that, although OPCs allow for a fast assessment of qualitative size information, the task to gain proper particle number size distributions can be challenging. One reason for this is the measurement principle itself, as particle size is only indirectly in-



ferred from scattered light intensity. This intensity, in general, is a non-monotonic function of particle size and depends also on particle intrinsic properties, such as complex refractive index and shape (Szymanski and Liu, 1986; Szymanski et al., 2009). Especially for particle sizes that are comparable or larger than the wavelength of the incident light the size-dependence of scattered intensity tends to be flat and occasionally ambiguous, so that uncertainties in the particle intrinsic properties can

introduce large sizing uncertainties (Reid et al., 2003; Formenti et al., 2011). Another reason lies in the existing methods for OPC calibration and response parametrization. The available approaches (e.g. Cerni, 1983; Bemer et al., 1990; Rosenberg et al., 2012; Cai et al., 2013) are not consistent with each other. Further, they do not allow for a comprehensive description of instrument response and a satisfactory quantification of corresponding uncertainties. Figure 1 summarizes the major sources of uncertainty adjunct to OPC measurements. They can be divided into the last-mentioned uncertainty in the instrument re-

sponse and calibration, the uncertainty in the particle intrinsic properties and the uncertainty in the measured concentrations themselves, e.g. arising from counting statistics, eventual particle losses etc. In order to allow for inter-comparability between different OPC instruments and the comparison with other measurement techniques it is necessary to correct for systematic errors and to quantify all uncertainties as good as possible, i.e. to improve OPC data accuracy and assess its precision (Formenti et al., 2011; Mahowald et al., 2014).

In the following manuscript we focus on the central aspect of OPC response modeling and calibration and present a new approach that

- allows for a more accurate description of OPC instrument response and

- yields realistic associated uncertainty estimates.

We discuss the advantages of the new approach against the background of the prevailing concepts and present its superiority
by means of measurement results for two optical particle counters that were used during the Saharan Aerosol Long-range TRansport and Aerosol-Cloud-Interaction Experiment (SALTRACE) (Weinzierl et al., 2016). Moreover, we outline a possible way to obtain adequate uncertainties for OPC size distributions within the new framework.

## 2   Methods

### 2.1   OPC Measurement Principle

The basic principle behind OPC measurements is that particles passing through a sample volume illuminated by a light source, usually a monochromatic laser, scatter light into a photosensitive detector. The amplitudes of the detected scattering signal pulses are a function of particle size. Counting the pulses and sorting them into discrete bins according to their amplitudes the measurement products of a typical OPC are scattering signal amplitude histograms. The mathematical problem of retrieving number size distributions from recorded scattering signal amplitude histograms is of inverse nature and is described by a set of





so called Fredholm integral equations of the first kind

$$N_i = \int\limits_0^\infty \kappa_i(D) F(D) dD \; (+\Delta N_i) \tag{1}$$

with the number of particles $N_i$ counted in bin $i$, a term $\Delta N_i$ accounting for potential counting errors, the corresponding kernel function $\kappa_i(D)$ giving the probability for each particle diameter $D$ to be sorted into bin $i$ and the number size distribution $F(D)$ (Kandlikar and Ramachandran, 1999; Fiebig et al., 2005).

Connecting the OPC output, i.e. the particle count histograms, and the desired information, i.e. particle number size distribution, the kernel functions are the key aspect of every OPC measurement. Deriving the kernel functions requires knowledge on the scattering signal amplitude threshold values defining the bin limits, the instrument-specific relationship between scattering signal amplitude and particle scattering cross-section and the theoretical relationship between scattering cross-section and particle size. The latter is subject to intrinsic particle properties such as complex refractive index and shape. For given size and intrinsic properties the particle scattering cross-section with respect to the incident light and OPC scattering geometry, i.e. the solid-angle range covered by the detector, can be calculated. In case of a homogeneous sphere Mie–Lorenz theory (Mie, 1908) provides an analytical solution. For more complex particle shapes complementary frameworks like the T-matrix method (Waterman, 1965) or the discrete dipole approximation (Purcell and Pennypacker, 1973) can be applied.

Bridging the gap between theoretical calculations and the instrument output, i.e. finding the instrument-specific parameters linking theoretical particle scattering cross-section and measured scattering signal amplitude is the purpose of an OPC calibration. The set of instrument-specific parameters resulting from the calibration in combination with scattering theory allows us to predict the OPC output, i.e. to determine the kernel functions, for any other material with given optical properties.

## 2.2 Existing Concepts for Size Assignment and Calibration Evaluation

Though scattering cross-section and, hence, signal amplitude generally is a non-monotonic function of particle size (see Fig. 2), the most popular approach of OPC bin size assignment is to assume or establish monotony in order to simplify Eq. (1) by allowing for a one-to-one mapping between particle diameter and bin threshold values. One way to achieve monotony is to replace the correct theoretical size dependence of the scattering cross-section by a smoothed monotonic approximation (Cerni, 1983; Osborne et al., 2008). Another option is to simply merge the bins in the affected size regions accepting a reduction in resolution (Pinnick et al., 1981). Following these concepts OPC manufacturers usually provide their instruments with a table of predefined (polystyrene latex equivalent) diameter bin threshold values. Mathematically, this means expressing the kernel functions as sharp, adjacent step functions in diameter space

$$\kappa_i(D) = \begin{cases} 1 & \text{for } D \in [D_i, D_{i+1}) \\ 0 & \text{otherwise} \end{cases}$$
$$= \int\limits_0^D \delta\left(\tilde{D} - D_i\right) - \delta\left(\tilde{D} - D_{i+1}\right) d\tilde{D} \tag{2}$$



with delta functions at $D_i$ and $D_{i+1}$, i.e. the lower and upper diameter threshold values of bin $i$. In doing so, Eq. (1) simplifies to

$$N_i = \int\limits_{D_i}^{D_{i+1}} F(D)\,dD$$

and the size distribution can be directly represented by the measured counts in a discrete way as

$$F(D) = \frac{N_i}{D_{i+1} - D_i} \text{ for } D \in [D_i, D_{i+1})$$

This simplification, however, involves fundamental shortcomings:

- Even if quasi-monotony in the theoretical scattering cross-section size dependence can be established for particles of certain intrinsic properties (e.g. polystyrene latex spheres) by a smart choice of OPC collecting optics (Barnard and Harrison, 1988) and/or bin threshold values, this does not automatically hold for particles of different intrinsic properties
(e.g. different refractive index or shape) (Szymanski et al., 2009).

- Due to the involved approximations (e.g. a smoothing of the theoretical scattering cross-section relationship) nominal manufacturer values can significantly deviate from reality for certain parts of the instrument size range. Such deviations are regularly reported (Szymanski and Liu, 1986; Rosenberg et al., 2012; Ryder et al., 2013).

- The instrument response can change over time, e.g. due to degradation of OPC light source intensity, pollution or mis-
alignment of optical elements. Such changes usually do not induce a uniform shift in the apparent size distributions, but rather cause a complicated deformation.

- No uncertainty estimates are provided for the nominal diameter threshold values. This lack entails an underestimation of size distribution uncertainties.

Some studies stick to the simplified concept of bin diameter threshold values but try to correct for possible diameter deviations.
Lance et al. (2010); Cai et al. (2013) use an empirical diameter offset to uniformly shift the manufacturer values in order to yield best agreement between measured histogram modes and nominal diameter values of reference particles. A more universal calibration approach commonly used is to find the parameters for the linear relationship between the measured (mean or mode) scattering signal amplitudes and theoretical (mean) scattering cross-sections for reference particles (Cerni, 1983; Bemer et al., 1990). Still assuming a monotonic relation between scattering cross-section and particle diameter they use the resulting linear
fit parameters (slope $m$ and intercept $c$) to derive the size dependence of the scattering signal amplitude and calculate the bin diameter threshold values from their predefined scattering signal amplitude counterparts.

Rosenberg et al. (2012) presented another way of size assignment avoiding workarounds for the non-monotonic behavior of particle scattering cross-section. Their main new concept is to use the linear fit parameters from the calibration and the unmodified theoretical relationship to define the kernel functions as diameter projections of the scattering signal amplitude



bins (see Fig. 2a)

$$\kappa_i(D) = \begin{cases} 1 & \text{for } C_{scat}(D) \in [C_{scat,i}, C_{scat,i+1}) \\ 0 & \text{otherwise} \end{cases}$$

$$= \begin{cases} 1 & \text{for } C_{scat}(D) \in \left[\frac{U_i-c}{m}, \frac{U_{i+1}-c}{m}\right) \\ 0 & \text{otherwise} \end{cases}$$

$C_{scat,i}$ and $C_{scat,i+1}$ denote the lower and upper scattering cross-section threshold values of bin $i$ that are a linear function of the actual thresholds given by the scattering signal amplitude values $U_i$ and $U_{i+1}$. This means, that particle diameters with an instrument-specific scattering cross-section of $C_{scat}(D)$ will be sorted into bin $i$ if $C_{scat}(D)$ falls within the limits defined by $U_i$ and $U_{i+1}$ scaled with the linear coefficients $m$ and $c$. This can be further expressed as

$$\kappa_i(D) = \int_0^D \delta\left(C_{scat}\left(\tilde{D}\right) - \frac{U_i-c}{m}\right) - \delta\left(C_{scat}\left(\tilde{D}\right) - \frac{U_{i+1}-c}{m}\right) d\tilde{D} \tag{3}$$

To simplify the inverse problem of Eq. (1) and, again, directly gain size distribution information from OPC histogram data they use the kernel functions to calculate so-called perfect (mean) diameters $D_{p,i}$ and widths $W_i$ to characterize all bins

$$D_{p,i} = \int_0^\infty D \cdot \kappa_i(D) \, dD$$

$$W_i = \int_0^\infty \kappa_i(D) \, dD$$

With these values a discrete representation for the size distribution is given by

$$F(D_{p,i}) = \frac{N_i}{W_i}$$

The uncertainties in the calibration parameters $m$ and $c$ are used to derive instrument-related uncertainties for $D_{p,i}$, $W_i$ and, therewith, the resulting size distribution values $F(D_{p,i})$. Although this approach supersedes workarounds for the ambiguities in the size dependence of the scattering cross-section it still has shortcomings. One conceptual inadequacy is that representing the bins by their perfect diameter $D_{p,i}$ is ultimately not appropriate, as it will only match with the real mean diameter of particles sorted into bin $i$ in the unrealistic case of a flat size distribution. If, for instance, the size distribution is (strongly) dropping towards larger particles, the occurrence of smaller particle diameters is more likely, meaning that the real mean diameter of particles falling into bin $i$ would be (much) smaller than $D_{p,i}$. As a result, this causes a sizing bias between the real and calculated size distribution.

Spiegel et al. (2012) offer yet another approach to directly estimate size distributions from measured histograms. They translate the range of possible scattering geometries seen by individual particles[1] into a range of possible particle diameter to

[1] In contrast to the instruments presented in this study, this is an important aspect for open path OPCs where particle positions with respect to the optics vary considerably.





scattering signal amplitude relationships, a so-called "Mie band". From this "Mie band" they calculate the a priori probabilities for discrete (equidistant) particle diameter intervals to contribute to the count rate of each bin. This approach potentially allows particle diameters to be sorted into more than one bin, i.e. overlapping OPC bins in diameter space. According to the derived contribution probabilities they then distribute the measured bin counts to the discrete diameter intervals. The major shortcoming

of this method is similar to the one discussed above. Even if the a priori probabilities for two diameter intervals (of equal width) to contribute to a certain bin's count rate are the same, the true particle abundance in these intervals will not be the same for the realistic case of a non-flat size distribution. Therefore, the inverse direction, i.e. to equally distribute the counted particles to the two intervals, is generally incorrect.

In summary, none of the existing concepts for OPC calibration and bin size assignment proofs completely satisfactory. The

simplifications to the inverse problem of Eq. (1) and the attempts to directly gain size information from OPC histogram data are always accompanied by systematic errors. Further, some approaches, especially the use of the manufacturer provided set of bin diameter threshold values, do not offer instrument-related (sizing) uncertainty estimates.

## 2.3 Instrumental Broadening of Size Spectra

A shortcoming common to all available methods is that they do not consider the artificial broadening of size distributions in

the basic parametrization of OPC response. The inhomogeneity of light intensity inside the sample volume (Wendisch et al., 1996) contributes significantly to this spectral broadening, i.e. the increase in apparent size distribution width. So far this has, if at all, been treated separately from the basic instrument calibration, although it is an instrument-specific property, meaning it is always present in OPC measurements.

Spectral broadening can be further enhanced by other effects such as varying orientation of aspherical particles with respect

to the direction of the incident light (Reid et al., 2003) and coincident count events (Baumgardner et al., 1985; Cooper, 1988). Latter becomes relevant for very high particle concentrations when average inter-particle distances are not larger than the size of the sample volume any more. In such a case, the probability to erroneously interpret the sum of several scattering signals from multiple particles as a single particle's signal increases. In addition to an artificial deformation of the size distribution towards larger sizes this entails an underestimation of total particle number concentration.

To correct for artificial spectral broadening the common procedure is to define a matrix that contains the probabilities (associated with the broadening effect) to find a particle of a certain size class in adjacent size classes in its elements (Cooper, 1988; Baumgardner and Spowart, 1990; Wendisch et al., 1996; Brenguier et al., 1998). The resulting inverse matrix equation is then solved for the true size distribution. One disadvantage of such methods based on empirical matrices is that their elements might not be universally valid, as for instance the magnitude of broadening that is related to varying particle orientations

depends on the degree of particle asphericity. Moreover, the number of uncertainty-afflicted parameters becomes quite large. Assuming an OPC with $K$ bins, the number of parameters required to describe spectral broadening is $K^2$.

Therefore, for the inversion of OPC histogram data it is advantageous to treat spectral broadening in a more universal way. In Sect. 2.5.1 we present a new approach that includes the instrument-specific part of spectral broadening within the basic



paramerization of OPC response. In addition, spectral broadening resulting from different orientations of aspherical particles can be included in the inversion process via a set of possible size to scattering cross-section relations as outlined in Sect. 2.5.3.

## 2.4 Uncertainty in Particle Properties

So far, we discussed the inverse nature of the OPC measurement principle, challenges and shortcomings in the parametrization of basic OPC response and the artificial broadening of size spectra. An aspect that further complicates OPC measurements is that, in most situations, the (size dependent) optical properties of the aerosol particles are a priori unknown or at least subject to a considerable degree of uncertainty. Externally or internally mixed individual particles can be combinations of different non-homogeneously distributed materials (e.g. Kandler et al., 2011) making it difficult to find representative complex refractive indices for the bulk aerosol. In any case, the quality of OPC derived size distributions depends on the quality of information on the optical particle properties.

In order to derive a size distribution uncertainty estimate from uncertainties in the particle properties, yet, most studies follow the pragmatic approach and report the maximum impact on the size distribution as a conservative estimate (e.g. Osborne et al., 2008; Weinzierl et al., 2009; Brock et al., 2011; Ryder et al., 2013; Hermann et al., 2016). However, the size distribution uncertainty induced by the uncertainties in the particle properties can be substantially size dependent. To yield improved size distribution uncertainty estimates one needs realistic estimates for the set of possible size to scattering cross-section relations and a proper way to propagate these estimates (as, for example, outlined in Sect. 2.5.3).

## 2.5 New Approach

In this section, we introduce an approach to the parametrization of OPC response that involves instrument-specific spectral broadening and overcomes the shortcomings of existing methods. We further propose a way to evaluate calibration measure-ments and to obtain aerosol particle number size distributions with realistic uncertainty estimates from OPC data.

### 2.5.1 Parametrization of the Instrument Response

Let a particle of intrinsic properties $\vartheta$ (complex refractive index etc.) and diameter $D$ have the scattering cross-section $C_{scat,\vartheta}$ with respect to the incident light and OPC scattering geometry. For an ideal instrument the scattering signal amplitude $U_\vartheta$ in the detector corresponding to this scattering cross-section would be given by the (assumed) linear relationship between scattering cross-section and scattering signal amplitude only, described by the coefficients for slope $m$ and intercept $c$. Due to the nature of real OPCs even spherical and homogeneous identical particles do not necessarily generate the same scattering signal amplitudes. To a large extent, this is expected due to the non-uniformity of light intensity in the sample volume. Particles passing at slightly different locations scatter light of different incident intensity which linearly confers on the intensity of the scattered light. As a consequence, the relative spread in scattering signal amplitudes can be assumed to be independent from their absolute value. Therefore, the single signal amplitude $U_\vartheta$ is replaced by a probability density function (PDF) for a set of possible $\{U_\vartheta\}$. In a simplified approach, this PDF can be assumed to follow a Gaussian distribution with a constant relative





standard deviation $b$. Equivalently, one could describe the initially sharp OPC scattering cross-section bin threshold values (resulting from the predefined scattering signal amplitude bin thresholds) as being "blurred" with the same relative standard deviation $b$. Replacing the delta functions in Eq. (3) (i.e. Gauss functions of vanishing standard deviation) by Gauss functions of constant relative standard deviation yields the new kernel functions

$$5 \quad \kappa_i\left(D \mid b, m, c\right) = \frac{1}{\sqrt{2\pi}b} \int_0^D \frac{1}{U_i} \exp\left(-\frac{\left(C_{scat,\vartheta}\left(\tilde{D}\right) - \frac{U_i - c}{m}\right)^2}{2b^2 U_i^2}\right) - \frac{1}{U_{i+1}} \exp\left(-\frac{\left(C_{scat,\vartheta}\left(\tilde{D}\right) - \frac{U_{i+1} - c}{m}\right)^2}{2b^2 U_{i+1}^2}\right) d\tilde{D} \qquad (4)$$

with the new instrument-specific parameter triplet $(b, m, c)$ and the scattering signal amplitude threshold values $U_i$ and $U_{i+1}$ defining bin $i$. Figure 2 illustrates the difference in OPC kernel functions between an ideal instrument that follows Eq. (3) and an instrument with a finite relative Gaussian broadening.

### 2.5.2 Calibration Evaluation

10   Including spectral broadening, the new parametrization allows for an extension of the classical OPC calibration evaluation approach, that is restricted to the determination of the linear coefficients for the relation between theoretical scattering cross-section and scattering signal amplitude.

Given a set of particle standards with known intrinsic properties and size distributions the forward solution of Eq. (1) using Eq. (4) for the kernel functions yields the model count histograms, i.e. the parametrized theoretical instrument response

$$15 \quad M_{ij} = \int_0^\infty \kappa_i\left(D \mid b, m, c\right) F_j\left(D\right) dD \qquad (5)$$

with the model counts $M_{ij}$ for OPC bin $i$ and particle standard $j$ and the corresponding number size distribution $F_j\left(D\right)$. With the real measured particle counts $N_{ij}$ the task of a calibration within the new framework is now to inversely find the values for the parameters that bring $M_{ij}$ and $N_{ij}$ into best agreement.

For stable measurement conditions, i.e. constant OPC volumetric sample flow etc., the uncertainties on the measured particle 20   counts follow the Poisson counting statistics. With increasing number of counts, the relative uncertainty hence decreases with $\sqrt{N_{ij}}$. Naturally, the simplified model will not be able to reproduce the calibration measurements perfectly, because there will be additional deviations that are not parametrized. Provided sufficiently high numbers of counts in the course of the sampling, the relative bin count uncertainties due to Poisson counting statistics will become negligible compared to these additional deviations. In consequence, bringing model and measurement into agreement corresponds to maximizing the probability for 25   the model counts $M_{ij}$ afflicted with unknown uncertainties $\sigma_{ij}$, that cover the additional model deviations, to occur given the measured $N_{ij}$. To ensure that the modeled instrument response later agrees with reality within its margin of uncertainty, it is necessary to find a good representation of the unknown uncertainties $\sigma_{ij}$ and quantify them in the course of the calibration, too.

Particle standard measurements show that the remaining deviations between modeled and measured OPC response mainly 30   appear as (non-uniform) small shifts of the histograms. It thus seems natural to treat these deviations as a remaining uncertainty





of modeled scattering signal amplitudes. Apart from the experimental finding there are also theoretical explanations for the observed shifts. Small differences between the actual OPC scattering geometry and the one used in theoretical calculations as studied by Rosenberg et al. (2012) can to first-order approximation be assumed to lead to a collective relative shift in theoretical particle scattering cross-sections. OPC light source intensity fluctuations can further cause non-collective relative histogram

shifts between samples that are recorded with certain time lags. These time-dependent intensity fluctuations can equivalently be thought of as relative fluctuations in the scattering cross-sections assuming a fixed intensity. Therefore, we can express the model count uncertainties $\sigma_{ij}$ in terms of a relative uncertainty of the theoretical particle scattering cross-sections $C_{scat,\vartheta}(D)$.

For a given instrument parameter tuple $(b,m,c)$ the set of model bin counts $M_{ij}$ results from Eq. (5) and (4) with the (best estimate) theoretical OPC scattering cross-section relation. A relative shift in the theoretical relationship corresponding

to a multiplication of $C_{scat,\vartheta}(D)$ by a factor $\varepsilon \neq 1$ leads to a different set of model bin counts $M_{ij,\varepsilon}$. Assuming the PDF of the possible relative shifts $\varepsilon$ to be a Gaussian function centered at 1 and having a standard deviation of $\sigma_\varepsilon$ one can derive the respective PDFs for the model bin counts $M_{ij,\varepsilon}$. For the sake of convenience and simplicity the resulting model bin count PDFs can themselves be approximated by Gaussian PDFs, which is usually an adequate approximation. This leads to the following expression for the unknown model bin count uncertainties

$$\sigma_{ij}^2 = \frac{1}{\sqrt{2\pi\sigma_\varepsilon^2}} \int_0^\infty (M_{ij,\varepsilon} - M_{ij})^2 \exp\left(-\frac{\varepsilon^2}{2\sigma_\varepsilon^2}\right) d\varepsilon \tag{6}$$

with $M_{ij,\varepsilon}$ defined by Eq. (5) and (4) replacing $C_{scat,\vartheta}(D)$ with $\varepsilon \cdot C_{scat,\vartheta}(D)$. In summary, the new calibration evaluation should yield the set of model parameters $(b,m,c,\sigma_\varepsilon)$ composed of the instrument-specific parameter tuple $(b,m,c)$ and, according to the above considerations, the remaining relative uncertainty of the theoretical scattering cross-section relation $\sigma_\varepsilon$.

A way to meet the challenge of model parameter probability maximization under initially unknown model uncertainties is to

make use of Bayesian statistics and Markov chain Monte Carlo (MCMC) methods (e.g. Goodman and Weare, 2010). Following Bayes' theorem (Bayes and Price, 1763) the (posterior) probability $P$ for a set of model bin counts $\{M_{ij}\}$ to occur under a set of measured bin counts $\{N_{ij}\}$ can be expressed as

$$P(\{M_{ij}\} \mid \{N_{ij}\}) \propto P(\{N_{ij}\} \mid \{M_{ij}\}) \cdot P(b,m,c,\sigma_{ij}) \tag{7}$$

, i.e. the product of the likelihood function determining the probability of the $\{N_{ij}\}$ to occur given the $\{M_{ij}\}$ and the so-called

prior probability $P(b,m,c,\sigma_{ij})$, including all prior knowledge on the model parameters for instance from physical constraints or invariance considerations (e.g. Jaynes, 1968). The proportionality factor equating both sides of Eq. (7) can be thought of as a normalization constant. Upon the assumption of Gaussian model bin count PDFs the likelihood function can be expressed as

$$P(\{N_{ij}\} \mid \{M_{ij}\}) = \prod_{ij} \frac{1}{\sqrt{2\pi\sigma_{ij}^2}} \exp\left(-\frac{(N_{ij} - M_{ij})^2}{2\sigma_{ij}^2}\right) \tag{8}$$

with $M_{ij}$ and $\sigma_{ij}$ defined by Eq. (5) and (6) respectively. MCMC methods allow us to efficiently sample the model parameter

space utilizing the forward solution to the problem to find the region of maximum probability according to Eq. (7). This way,





the PDFs for the instrument parameters $(b, m, c)$ and the relative uncertainty parameter for the theoretical particle scattering cross-sections $\sigma_\varepsilon$ are obtained together with all correlations between the individual parameters. In this study we utilize the Python-based sampler tool *emcee* (Foreman-Mackey et al., 2013).

### 2.5.3    Retrieval of Size Distributions within the New Framework

The new instrument parametrization including instrument-specific spectral broadening and the parameter PDFs resulting from the MCMC-based calibration evaluation now enable us to derive OPC size distributions in a self-consistent way. Propagating the parameter uncertainties yields improved estimates for the corresponding size distribution uncertainties for arbitrary ambient aerosol. Figure 3 illustrates a possible workflow within the proposed framework to go from measured OPC count histogram data to PDFs in size distribution solutions. Similar to what has been proposed by Fiebig et al. (2005) the basic idea is to start

with random Monte Carlo samples drawn from the model parameter PDFs and a set of possible theoretical particle diameter to scattering cross-section relationships $\{C_{scat,\vartheta}(D)\}$, e.g. given by the likely range of aerosol particle complex refractive indices, their shape and orientation[2]. In addition, a random relative shift from the chosen $C_{scat,\vartheta}(D)$ is picked according to its possible systematic deviation from the real relationship. This relative shift is drawn from the PDF parametrized by $\sigma_\varepsilon$, which is derived as part of the calibration evaluation. With the resulting diameter to scattering cross-section relation and the

instrument parameter tuple the set of OPC kernel functions can be calculated following Eq. (4). By adjusting the aerosol size distribution under this set of bin kernel functions to minimize the deviation between model and measured histograms given the measurement uncertainties the result is then either one best solution for the size distribution or an ensemble of possible solutions for each iteration, depending on the respective inversion algorithm (see e.g. Kandlikar and Ramachandran, 1999; Fiebig et al., 2005, and the references herein). By repeating this procedure multiple times one finally acquires a collective PDF for the size

distribution solutions considering all uncertainties in instrument specific parameters and the theoretical diameter to scattering cross-section relations. In this work (see Sect. 4) we use a parametrized size distribution and, again, a MCMC method to obtain corresponding parameter PDFs for each Monte Carlo iteration, which are then merged into final size distribution parameter PDFs.

## 3    Experimental Setup

### 3.1    Involved OPCs

The two central OPCs examined in this study are the Grimm model 1.129 (SkyOPC) and the DMT Airborne Passive Cavity Aerosol Spectrometer Probe (PCASP-100X with an upgraded signal processing package SPP-200, abbreviated PCASP hereafter). Both aerosol spectrometers were part of the airborne in situ instrumentation used in the SALTRACE campaign. The SkyOPCs were operated inside the cabin of the German Aerospace Center's Falcon research aircraft behind an isokinetic

aerosol inlet, the PCASP was mounted in one of the under-wing stations. Detailed descriptions of the instruments can be found

---

[2]Other uncertainty-afflicted instrument properties as, for instance, sample flow rate or size-dependent aspiration efficiency can be randomly sampled in a comparable manner.



in Bundke et al. (2015) for the SkyOPC and in Liu et al. (1992) and Strapp et al. (1992) for the PCASP. Both are closed path spectrometers in which the aerosol particle beam is confined to the inner area of light source focus. The particles scatter light coming from a monochromatic laser of visible red wavelength (633nm Helium-Neon laser for the PCASP and 655nm diode laser for the SkyOPC) which is then detected in a sideways direction. The applied wide-angle collection of scattered light minimizes ambiguities in the particle size to scattering cross-section relationship (see Heim et al. (2008) and Rosenberg et al. (2012) for details on the scattering geometry).

During SALTRACE and the lab measurements presented here the SkyOPC was operated in the fast mode for smaller sizes covering a nominal diameter range of 0.25 to about $3\mu$m. The corresponding scattering signal amplitude range is separated into 16 preset bins defined by set of digital threshold values. In standard configuration, the PCASP sorts scattering pulses into 30 bins over a nominal diameter range of 0.1 to $3\mu$m. It further allows for a custom selection of the digital bin threshold values. In this study, we only consider the PCASP low gain stage[3]. In order to better study differences between the approaches discussed in Sect. 2, we present results for a custom high resolution binning of this gain stage in addition to the default binning. For the custom binning the gain stage's signal amplitude range is divided into bins with constant width.

The DMT Ultra-High Sensitivity Aerosol Spectrometer (UHSAS) (Cai et al., 2008) lab version covering a size range of about 0.06 to $1\mu$m in high resolution (99 bins) was utilized as a reference for total particle concentration during the SkyOPC calibration measurements and further served for a qualitative assessment of sizing to support the particle mobility filtering described in Sect. 3.2.

## 3.2 Measurements

The calibration measurements were performed using monodisperse aerosols of polystyrene latex (PSL) spheres. For the SkyOPC, the data set is complemented by di(2-ethylhexyl) sebacate (DEHS) aerosol samples. The complex refractive indices for PSL and DEHS are approximately $1.585 + i0$ (Sultanova et al., 2009) and $1.45 + i0$ (manufacturer data sheet) in the wavelength range of the SkyOPC and PCASP. PSL spheres dispersed in distilled water were mobilized via nebulization with the DMT portable aerosol generator running with aerosol-free carrier air. The resulting aerosol was subsequently dehumidified by an arrangement of silica gel dryers and diluted to avoid OPC measurement issues related to coincident count events. The DEHS aerosol was produced using a TSI Model 3475 condensation aerosol generator based on the Sinclair-LaMer principle (Altmann and Peters, 1992): with nitrogen as the carrier gas an aqueous sodium chloride solution is nebulized and dried to yield a high-concentration condensation nuclei aerosol. Passing through a heated vessel filled with liquid DEHS, a reheater unit and a condensation chimney the precursor particles allow for heterogeneous condensation of the supersaturated DEHS vapor. The mean particle size of the resulting (quasi-)monodisperse aerosol is a function of the ratio between vapor concentration and condensation nuclei number concentration. Again, the DEHS aerosol was diluted prior to the measurement in the OPC to circumvent counting coincidences. For all measurements we chose sampling interval times long enough to minimize relative uncertainties from counting statistics.

---

[3]As the intensity of scattered light intensity over the PCASP size range covers more than six orders of magnitude the PCASP optical detection system is divided into three amplification stages, called the high gain, mid gain, and low gain stage.



For mean particle diameters up to 800nm the aerosol was additionally filtered with the aid of a differential mobility analyzer (Grimm Vienna type L-DMA, abbreviated DMA hereafter, Reischl et al. (1997)). DMA filtering substantially reduced the widths of the DEHS aerosol size distributions and allows us to obtain quantitative information on their mean diameters and widths. Using the UHSAS, it was taken care that the DMA transfer function was centered to the middle of the initial DEHS

generator aerosol size distribution to guarantee that the resulting size distribution is in good approximation represented by the Gaussian DMA transfer function itself. The relative standard deviations of the DMA transfer functions are calculated by means of the formulae given in Reischl et al. (1997) and Stolzenburg (1988). The initial PSL particle size distributions, that are traceable via the United States National Institute of Standards and Technology (NIST), are of Gaussian shape with known mean and standard deviation. Although they are narrower than the width of the DMA transfer functions, additional DMA

filtering helped to effectively remove the interfering background at smaller particle diameters that is caused by the nebulization (cf. Hermann et al. (2016)). For mean particle diameters larger than 800nm the presented counting histograms are empirically separated from this background.

## 4   Results and Discussion

In this section, we present the results for the evaluation of the PSL calibration measurements following the new method

proposed in Sect. 2.5. We compare these results with the theoretical instrument response for nominal manufacturer diameter bin thresholds and results obtained for the approach of Rosenberg et al. (2012), representing a state-of-the-art conventional method. Hereafter, we abbreviate these approaches as MFR and R12, respectively. We further demonstrate the impact of method choice on size distribution inversion results for measurements of DEHS samples.

The measurements of PSL particle standards, carried out as described in Sect. 3.2, are utilized to calibrate the OPCs following

both the new and the R12 approach (introduced in Sect. 2.2). Figure 4 and 5 contrast the resulting modeled relative bin count histograms and the measured relative histograms for the SkyOPC and the PCASP (low gain stage) respectively. The model histograms are calculated by means of Eq. (5) with the well-defined Gaussian PSL size distributions and the kernel functions given by Eq. (3) for the R12 (shown in red brown colors) and Eq. (4) for the new approach (shown in blue colors). The best estimate model histograms, i.e. the model histograms for the maximum probability model parameter tuple — $(m,c)_{best}$ for the

R12 and $(b,m,c,\sigma_\varepsilon)_{best}$ for the new approach — are represented by the color-framed white histogram bars. For the SkyOPC, additionally the model histograms for the MFR approach following Eq. (2) and using the manufacturer-supplied set of nominal values are displayed in golden colors. The underlying measured histograms are depicted by the gray bars. For the new and the R12 approach the parameter PDFs resulting from the evaluation of the calibration measurements (see Fig. A1 and A2) are sampled using a Monte Carlo method to yield the corresponding PDFs of the model histogram bin counts that are visualized

by error bars spanning the range between the 16 and 84th percentiles. In each panel of Fig. 4 and 5 the mean diameter and standard deviation of the Gaussian PSL size distribution is displayed in the left upper corner. Figure 6 supplements the SkyOPC histogram comparisons with scatter plots showing all modeled and measured relative bin counts for the different approaches.





Finally, Fig. 7 quantitatively compares the total sum of residuals

$$\sum_{ij} R_{ij,best} = \sum_{ij} \sqrt{\left(N_{ij} - M_{ij,best}\right)^2}$$

between the measured $N$ and best estimate model relative bin counts $M$ for the two instruments and the different approaches. The subscripts $i$ and $j$ represent the different OPC bins and used particle standards respectively.

The model histograms for the MFR approach (e.g. Fig. 4, golden colors) exhibit significant deviations from the underlying measured histograms. They offer much smaller widths than their measured counterparts. In addition, absolute offsets between the histogram modes are apparent for both SkyOPC and PCASP (not shown). Deviations are largest for the SkyOPC, because it was operating under dusty conditions during SALTRACE over a longer period previous to the presented measurements, presumably causing a pollution of optical elements. In consequence, the scatter plots for the MFR approach in the upper row of

Fig. 6 show the largest discrepancy between model and measurements. This becomes also obvious for both instruments when looking at the total sums of residuals in Fig. 7. The residuals for the MFR approach are substantially enhanced compared to the others.

     The R12 approach allows for the correction of the absolute shifts of the histogram modes. Nevertheless, instrument-specific spectral broadening is still ignored. The modeled histograms, thus, continue to underestimate the widths of the actually mea-

sured histograms, which is visible in the histogram plots in Fig. 4 and 5. Here, and especially in Fig. 6 it is also apparent that the R12 approach remains unable to reproduce the measurements within the margins of model uncertainty for most of the relative bin counts. Particularly for the smaller relative count values the absence of a parametrization of spectral broadening leads to large model deviations. However, in comparison to the MFR approach total residuals for the model best estimates are reduced by 25 and 35% for the SkyOPC and PCASP (low gain stage, default binning) respectively. Beyond that, an estimate

for the model uncertainty is established.

     By introducing a simple parametrization of instrument-specific spectral broadening and a self-consistent way of evaluating OPC calibration measurements, the new method succeeds in modeling the measured histogram widths correctly (see Fig. 4 and 5 rightmost columns). As a result, the total residuals between measured and modeled relative bin counts for the model best estimates decrease by 82 and 77% compared to the MFR approach for the SkyOPC and PCASP (low gain stage, default

binning) respectively. With respect to the R12 approach total residuals for the SkyOPC, the PCASP default binning and the finer PCASP custom binning are lowered by 77, 64 and 76%. Further, Fig. 6 shows that the new approach proofs capable to correctly reproduce the measured histograms within the margins of model uncertainty over the complete range of relative bin counts.

     Figure 8 shows the SkyOPC counting efficiency curve obtained by parallel measurements with the UHSAS as a reference

counter during the PSL calibration measurements. These measurements offer another perspective on the comparison between the two approaches. The mean total concentration fractions measured by the SkyOPC

$$f_{msm,j}\left(D_j\right) = \frac{\sum_i N_{ij,\mathrm{SkyOPC}}}{\sum_k N_{kj,\mathrm{UHSAS}}} \cdot \frac{\varPhi_{\mathrm{UHSAS}}}{\varPhi_{\mathrm{SkyOPC}}}$$





are calculated from the respective total number of counts and the volumetric instrument flow rates $\Phi$ for each particle standard $j$ and are depicted by the red diamond markers. The associated 68% confidence intervals (approximately corresponding to $\pm$ one standard deviation) that result from error propagation involving count rate scatter and instrument sample flow uncertainties are represented by the red error bars. The modeled concentration fractions are derived from the bin kernel functions $\kappa_i$ as

$$5 \quad f_{mdl}(D) = \sum_i \kappa_i(D)$$

and are visualized by the solid lines for the model best estimates, again in red brown for the R12 and in blue for the new approach. The shaded areas show the range between the 16 and 84th percentiles derived from the model parameter PDFs. The R12 approach predicts a sharp drop-off to smaller particle diameters in contrast to the measurements. The new approach is able to correctly model both shape and absolute values of the observed sigmoidal behavior of the counting efficiency curve.

10    Measurements of DEHS samples, as outlined in Sect. 3.2, allow us to test the possible implication of the choice of method for size distribution inversion results using an independent material. As proposed in Sect. 2.5.3 and illustrated in Fig. 3, the inversion of measured OPC histogram data is based on the parametrization of instrument response, the respective parameter PDFs derived from the calibration and the particle diameter to scattering cross-section relationship for the new material. The use of DEHS spherical droplets guarantees that this latter relationship is well-defined for the given scattering geometry as complex refractive index and shape of the aerosol particles are known, thus adding no further complexity to the retrieval. Moreover, the size distribution of the filtered DEHS samples follows a Gaussian distribution simplifying the inversion in this case to the determination of the size distribution parameters, mean diameter $\mu_{sd}$ and standard deviation $\sigma_{sd}$. The inversion algorithm used here to solve Eq. (1) for the parametrized size distribution is based on a MCMC method (Goodman and Weare, 2010; Foreman-Mackey et al., 2013). To obtain adequate size distribution parameter PDFs 10000 Monte Carlo samples are drawn from the corresponding instrument parameter PDFs. Figure 9 shows the inversion results for two DEHS samples and the two methods, i.e. the R12 approach (red brown) and the new one (blue). The theoretical (true) values for the size distribution means and standard deviations are depicted by the red markers and lines. In all cases shown the retrieved size distribution means agree with the theoretical values within their range of uncertainty, meaning that both methods allow for a correct (mean) sizing. This finding additionally proofs the validity of the used particle diameter to scattering cross-section relationships. Upon closer inspection the retrieved means tend to slightly underestimate the true values, which could imply minor deviations between the true and the used OPC scattering geometry and/or refractive index values for PSL and DEHS. The parameter PDFs for the size distribution means $\mu_{sd}$ are almost identical for the two methods concerning both, PDF median values and widths, i.e. uncertainty ranges. This agreement disappears for the size distribution standard deviations. The new method again agrees with the theoretical values within the range of parameter uncertainty and, hence, successfully predicts the full shape of the size distribution. The R12 approach attributes the spectral width of a measured histogram completely to the width of the size distribution, thus overestimating this width significantly. For the examples shown here the theoretical standard deviation values are overestimated by 714 and 302% with respect to the medians of the retrieved parameter PDFs. The widths of the $\sigma_{sd}$ parameter PDFs, i.e. the estimated range of uncertainty in this parameter, also differs for the two methods. With respect to the distance between 16 and 84th percentiles the R12 approach yields 285 and 224% higher PDF widths than the new method



leading to greater overall uncertainties of the retrieved size distributions, which are nonetheless unable to encompass the true ones.

It should be noted, though, that the standard deviations of the DEHS size distributions are quite small. When size distributions become broader the impact of instrument-specific spectral broadening on the width of the recorded histograms decreases and, hence, differences between the methods will become less pronounced. Besides, uncertainties in aerosol properties like complex refractive index and shape might be the dominant source of size distribution uncertainty in many situations. However, this example demonstrates that the new method is able to retrieve even narrow size distributions correctly and, hence, to provide access to realistic uncertainty estimates for all situations. The results also imply that even for the same data and OPC instrument, calibrated with the same set of measurements, retrieved size distributions can be contradictory solely due to different instrument response parametrizations and calibration evaluation approaches.

## 5 Conclusions

Retrieving aerosol particle number size distributions and associated uncertainties from OPC histogram data is a challenging task. Scattered light intensity (the measurand) generally is a non-monotonic function of particle size (the quantity of interest) and depends also on particle intrinsic properties such as complex refractive index. Besides, due to the non-ideal behavior of real OPCs, measured intensity distributions are artificially broadened. To realistically model OPC response, i.e. to find suitable OPC bin kernel functions defining the probabilities for particle diameters to be sorted into the instrument's discrete scattering signal amplitude bins, hence, is a crucial requirement.

We have introduced a new approach to model OPC response and, within this framework, a self-consistent way for the evaluation of calibration measurements. Two OPCs involved in the SALTRACE campaign, the SkyOPC and the PCASP, and measurements of PSL particles have been utilized to compare the new approach with existing concepts. The results lead to the following conclusions:

The manufacturer provided set of (PSL equivalent) nominal diameter threshold values for the OPC bin borders should be treated with caution and the resultant size distributions should be considered as rather qualitative measures. Not only the concept of adjacent continuous bins in diameter space can be problematic given the non-monotonic relation between particle size and scattering signal amplitude, but the values are also material-dependent and drifts in size-assignment, e.g. due to pollution of OPC optics or light source intensity drifts, can occur over time. We have shown that the resultant size distributions can significantly deviate from reality, even for the reference material. Furthermore, no uncertainty estimates are provided for the nominal diameter values that could be used to infer instrument-related size distribution uncertainties.

Calibrating the instrument can remove absolute sizing offsets. The results for a state-of-the-art OPC calibration and response parametrization approach (Rosenberg et al., 2012) exhibit clear improvements in sizing and, therewith, a reduction in total residuals between modeled and measured bin histograms. The introduction of instrument parameter uncertainties that goes along with the calibration evaluation allows to derive size distribution uncertainty estimates. However, these estimates fail to



explain remaining differences between modeled and measured instrument response for the presented data. The main reason for this is the absence of a parametrization of instrument-specific spectral broadening.

By introducing a simple (one parameter) approach to describe this ever-present broadening of size spectra the new method leads to substantial improvements. Residuals between modeled and measured OPC response are considerably reduced com-
pared to the other methods. The new method further correctly predicts the size-dependence of OPC counting efficiency. Most importantly, the measurements are successfully reproduced within the range of model uncertainty.

In the context of the new method we have also outlined a self-consistent way to propagate resulting uncertainties and gain size distribution PDFs without avoiding to address the actual inverse problem underlying OPC measurements. Exemplary inversion results for measurements of DEHS samples demonstrate the new method's ability to correctly retrieve (even narrow) size
distributions by combining the instrument parametrization with theoretical scattering calculations, whereas the conventional method fails to model the size distribution widths properly.

In summary, the new method has the following major advantages over existing concepts for OPC bin size assignment:

- The inevitable instrument-specific broadening of measured size spectra is parametrized for the first time leading to a more accurate modeling of OPC response.

- The model parameter PDFs resulting from the evaluation of calibration measurements allow for realistic uncertainty estimates for this response and, in consequence, provide a basis for proper size distribution uncertainties.

**Appendix A**

Figure A1 and A2 show the calibration results for the PCASP low gain stage with the custom high resolution binning following the new and the R12 approach, respectively. As explained in detail in Sect. 2.5 the parameter solution ensemble for the new
approach is obtained with the aid of the Python-based MCMC sampler tool *emcee* (Foreman-Mackey et al., 2013) using Eq. (8) for the likelihood function.

For the R12 approach the theoretical particle scattering cross-section means $\overline{C_{scat\,j}}$ and standard deviations $(\sigma_{C_{scat}})_j$ for each particle standard $j$ are calculated from the known PSL size distributions and the theoretical size dependence of the scattering cross-section following Eq. (4) and (5) in Rosenberg et al. (2012). Accordingly, for the scattering signal amplitudes the
measured histogram mode bin mids $\overline{U}_j$ and half widths $(\sigma_U)_j$ are calculated. Further adapting their procedure, for the linear fit between the two properties the scattering cross-section and signal amplitude PDFs for each particle standard are approximated by uncorrelated Gaussian distributions with the afore-mentioned values defining the respective means and standard deviations. In order to yield a parameter solution ensemble similar to the new approach, in this study the fitting is conducted by means of the same MCMC tool using the following logarithmic likelihood function for a linear relation between two properties with
uncorrelated Gaussian uncertainties:

$$\ln P = K - \sum_j \frac{\left(m \cdot \overline{C_{scat\,j}} + c - \overline{U}_j\right)^2}{2\left(m^2 \cdot (\sigma_{C_{scat}})_j^2 + (\sigma_U)_j^2\right)}$$





$m$ and $c$ represent the fit slope and intercept, $K$ is a (neglectable) constant offset that has no influence on the maximization process and the resulting parameter solution PDFs.

The two approaches' results for $m$ and $c$ are consistent with respect to the corresponding uncertainty ranges (see Fig. A1 and A2) with the new method yielding smaller uncertainty ranges for both $m$ and $c$. The median values for the new method's additional parameters $b$ and $\sigma_\varepsilon$, describing the relative standard deviation ("blurring") of the scattering cross-section bin threshold values and the remaining relative uncertainty of the theoretical scattering cross-section relation, approximately amount to 22 and 10%, respectively. The slight cutting of larger values in the PDF for $c$ resulting for the R12 approach is caused by the physical constraint that the scattering cross-section values for the bin thresholds may not be negative.

*Acknowledgements.* The research leading to these results has received funding from the Helmholtz Association under grant number VH-NG-606 (Helmholtz-Hochschul-Nachwuchsforschergruppe AerCARE) and from the European Research Council under the European Community's Horizon 2020 research and innovation framework program/ERC grant agreement number 640458 (A-LIFE). The SALTRACE campaign was mainly funded by the Helmholtz Association, DLR, TROPOS, and LMU. We further acknowledge funding from the LMU Munich's Institutional Strategy LMUexcellent within the framework of the German Excellence Initiative, and from the European Union through the European Seventh Framework Programme (FP7 2007-2013) under grant agreement number 607905 (Marie Curie Initial Training Network VERTIGO). We also would like to thank M. Richter (GRIMM Aerosol Technik GmbH & Co. KG, Ainring, Germany) for fruitful discussions and information about their instruments. We acknowledge the use of the software module providing the corner plots by Foreman-Mackey et al. (2016).



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





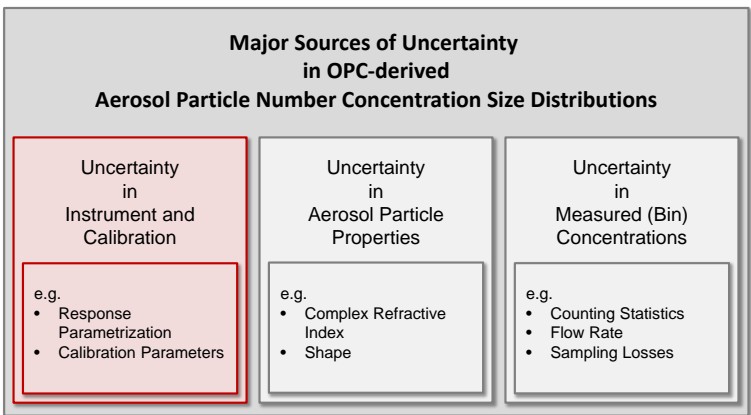

**Figure 1.** Sources of uncertainty associated with size distributions derived from OPC measurements.



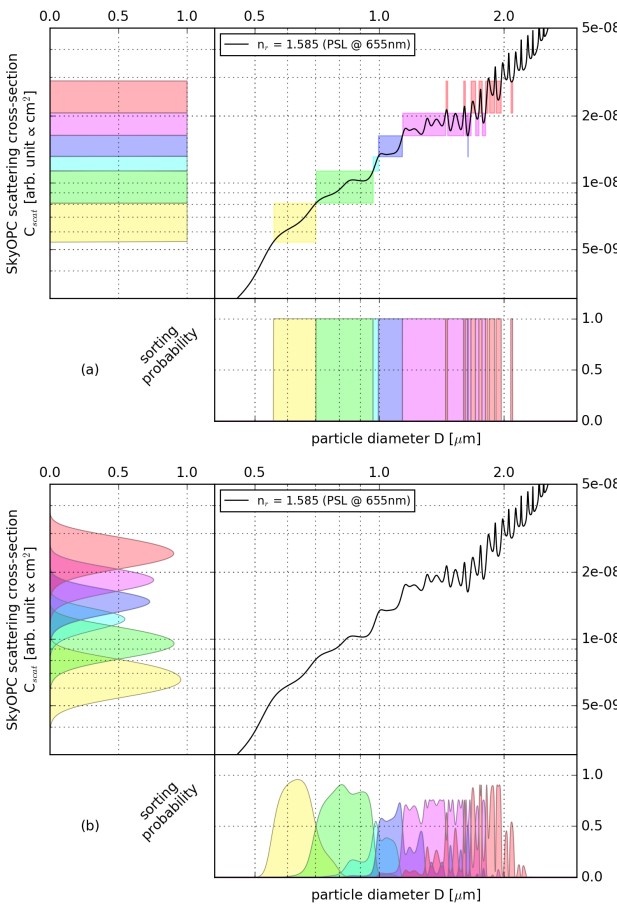

**Figure 2.** An example subset of kernel functions for the SkyOPC describing the probabilities for particle diameters and corresponding scattering cross-sections to be sorted into the predefined OPC scattering signal amplitude histogram bins, visualized by the different colors. The theoretical relationship between particle diameter and scattering cross-section for PSL is represented by the black curve. The upper graph (a) shows an ideal case without instrumental broadening of size spectra, whereas the lower graph (b) shows a more realistic case where the effect of spectral broadening is considered. The broadening is parametrized by a constant relative Gaussian uncertainty on the scattering cross-section bin threshold values.





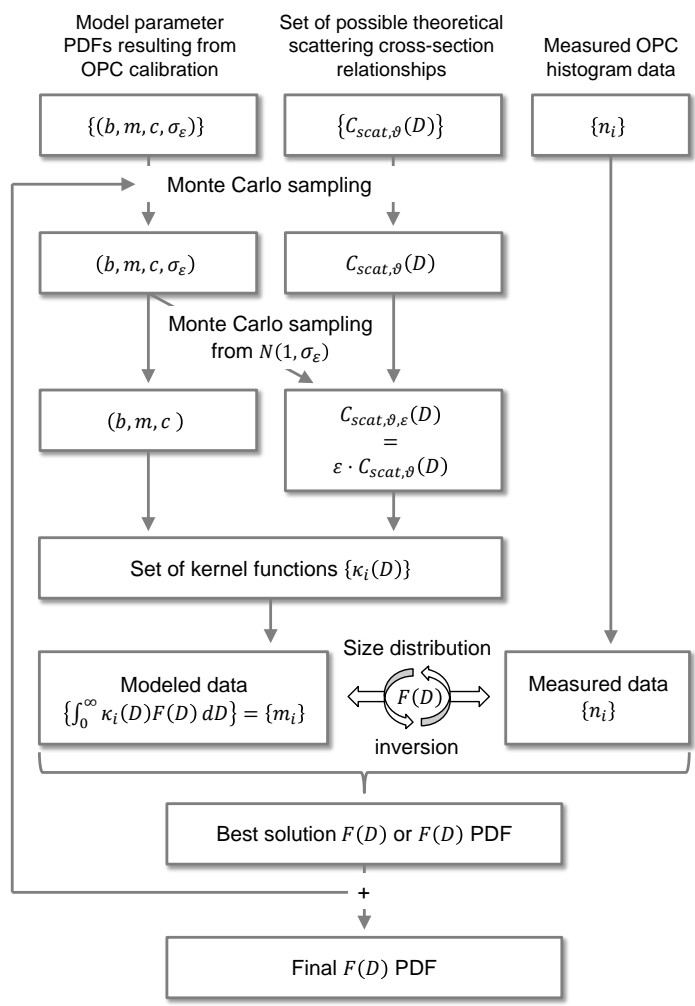

**Figure 3.** Flow chart demonstrating a possible pathway for the retrieval of size distribution information from OPC histogram data within the new framework.





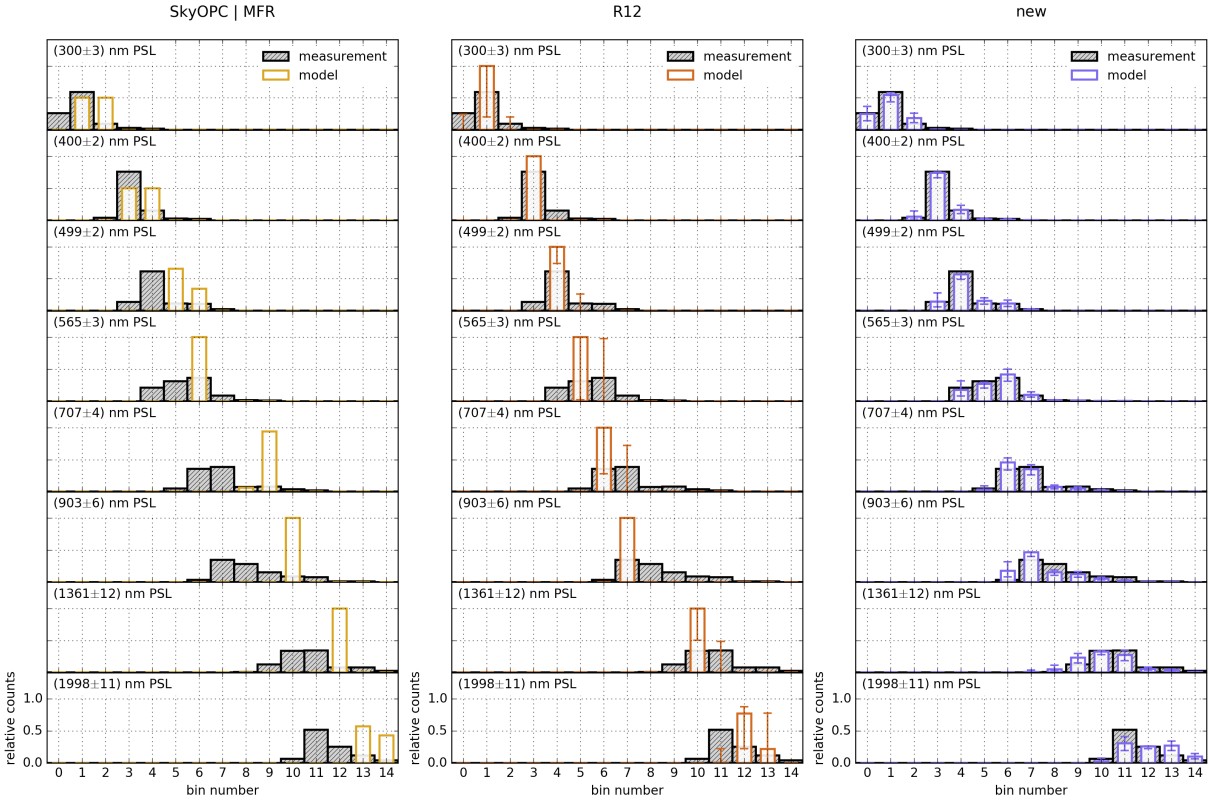

**Figure 4.** Comparison of modeled relative histograms (colored) and measured counterparts (gray, hatched) for the SkyOPC and different PSL particle standards (rows) for different approaches of OPC kernel function parametrization (columns). The colored histogram bars represent each model's best estimate, the error bars the range between the 16 and 84th percentiles of the corresponding PDFs. The leftmost column shows the theoretical instrument response according to the manufacturer provided set of nominal diameter threshold values in gold (MFR), the middle column the results following the calibration and instrument parametrization approach by Rosenberg et al. (2012) in red brown (R12) and the rightmost column the results of the new approach in blue.



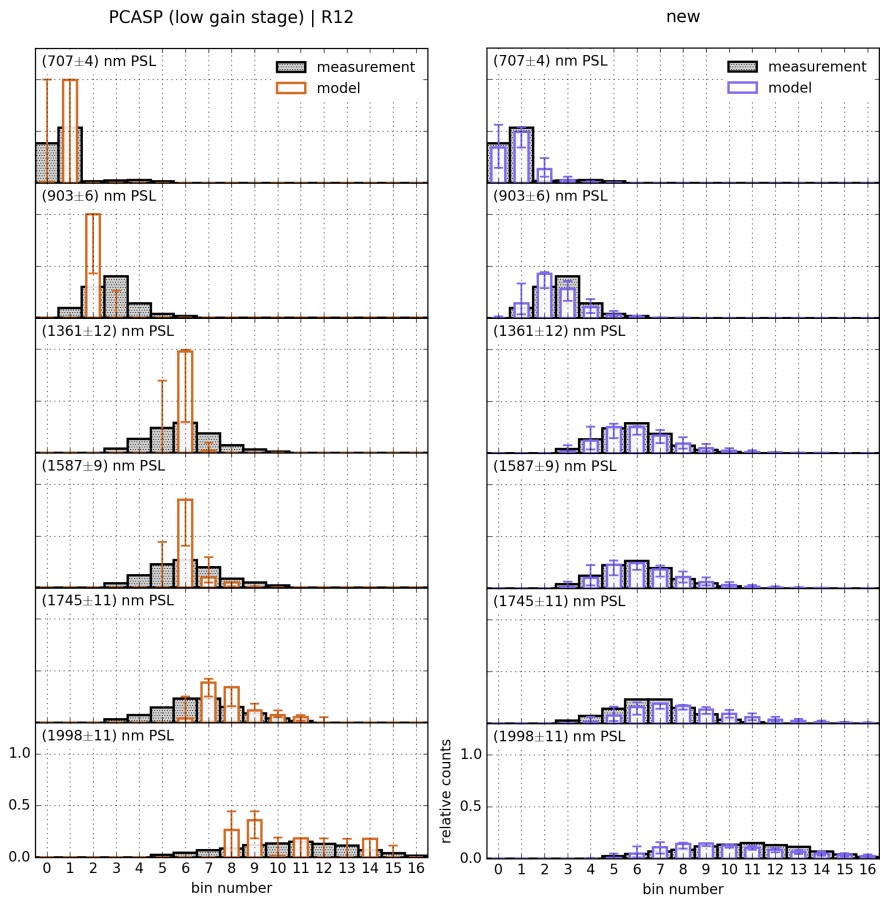

**Figure 5.** Comparison of modeled relative histograms (colored) and measured counterparts (gray, dotted) for the PCASP and different PSL particle standards (rows) for different approaches of OPC kernel function parametrization (columns), by analogy with Fig. 4. Instead of the PCASP (low gain stage) default binning a custom high resolution linear partitioning is applied here to better highlight the differences between the approaches.





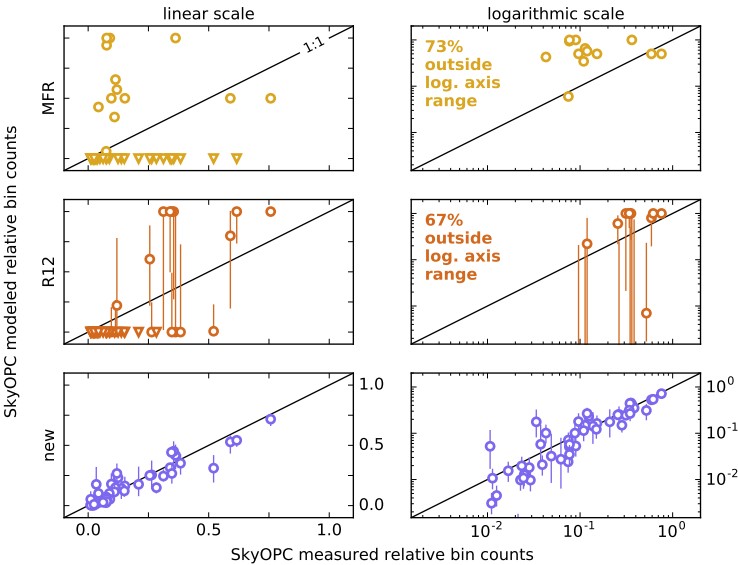

**Figure 6.** All modeled relative SkyOPC bin counts for the PSL standards scatter-plotted versus their measured counterparts for the three different approaches (rows). The comparisons are shown on linear and logarithmic scales on the left and right hand side respectively. The markers represent the model best estimates, the error bars the range between the 16 and 84th percentiles of the corresponding PDFs. The black lines follow the one-to-one relationship. Significant model underestimations, i.e. vanishingly small model values where non-vanishing bin counts are measured, occur in the two upper rows. The number fraction of significantly underestimated values is noted in the upper left corner of the logarithmic scale plots and the corresponding values are shown with triangular markers in the linear scale plots.



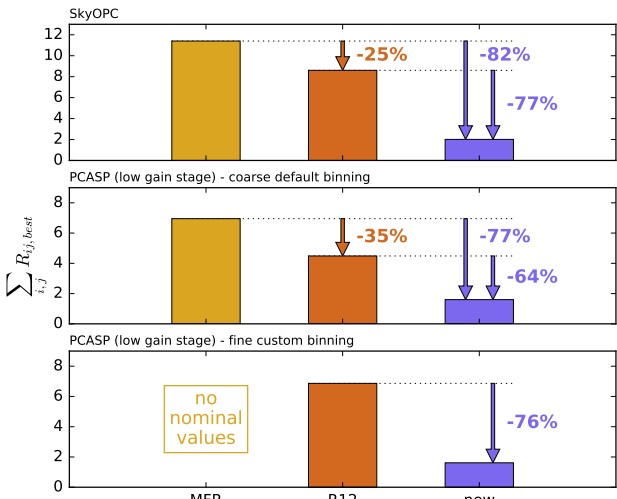

**Figure 7.** Total sum of residuals between measured relative bin counts and the corresponding model best estimates including all PSL calibration measurements for the SkyOPC and PCASP. In addition to the absolute residual values (solid bars), the arrows and percentage numbers demonstrate the relative reduction by changing the approach.

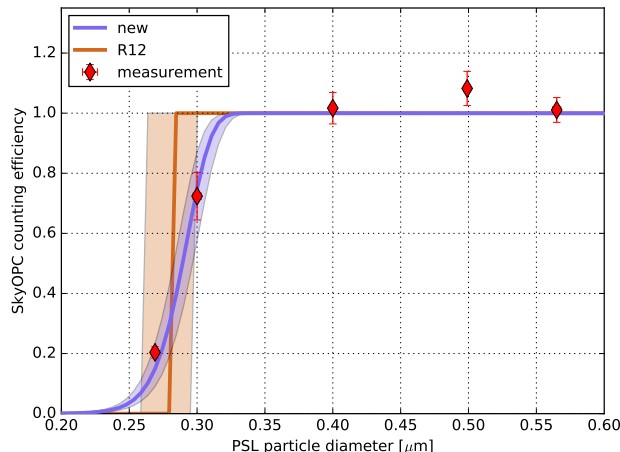

**Figure 8.** Comparison between modeled and measured SkyOPC (bin 1-15) counting efficiency. The measured mean counting efficiency values are plotted with red diamond markers and their associated 68% confidence intervals with red error bars. The solid lines represent the model best estimates for the different approaches. The shaded areas correspond to the range between the 16 and 84th percentiles.





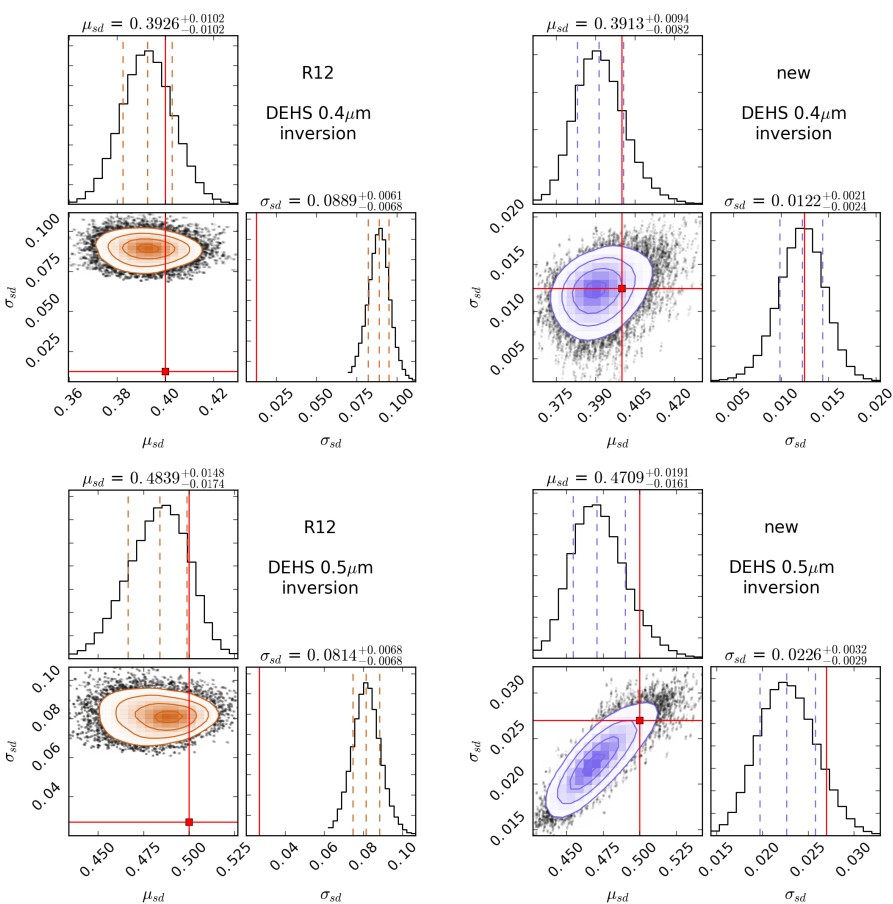

**Figure 9.** Parametrized size distribution retrieval results for two DEHS samples with mean diameters of 0.4 (upper row) and 0.5$\mu$m (lower row) using the different approaches (columns). The theoretical (true) values for the Gaussian aerosol distributions' means and standard deviations are indicated by the red lines and markers. The dashed lines in the 1D parameter solution histograms represent the median, 16 and 84th percentiles (in $\mu$m). The median values and their distances to the percentiles are noted on top of each histogram. The 2D plots show the solution scatter (black points) superposed with color-coded 2D histograms and smoothed Gaussian contours at 0.5, 1., 1.5 and 2 sigma.





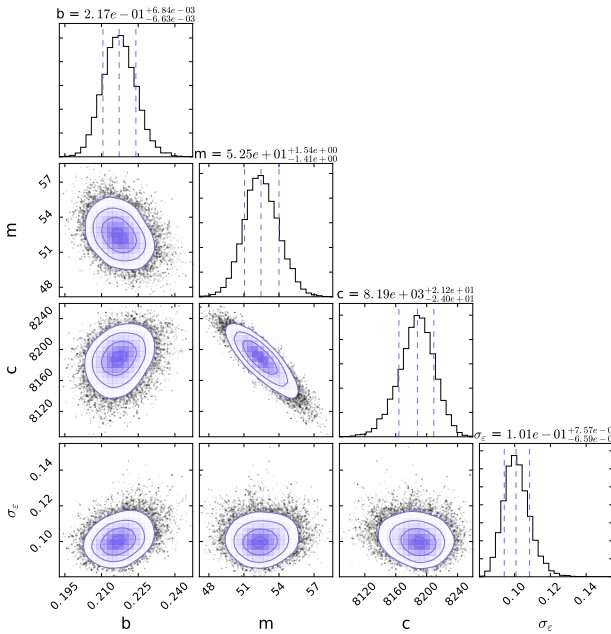

**Figure A1.** PCASP (low gain stage) calibration results following the new method. Shown is the resulting model parameter ensemble by analogy with Fig. 9.

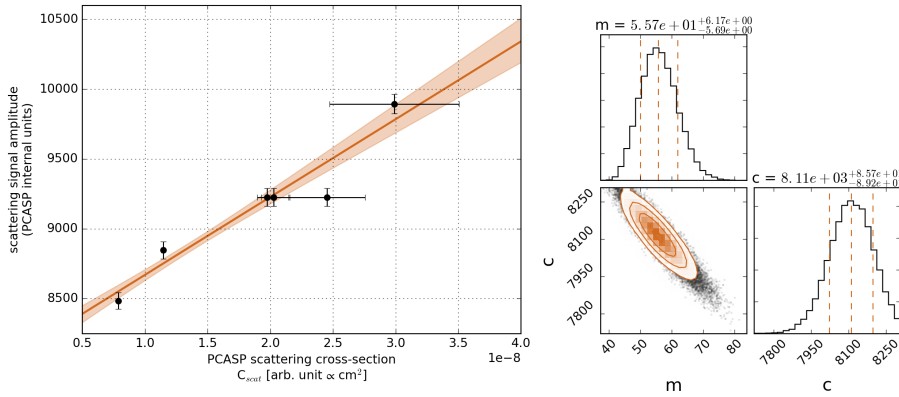

**Figure A2.** PCASP (low gain stage) calibration results for the approach of Rosenberg et al. (2012). The black markers in the left hand plot represent the mids of the scattering signal amplitude histogram modes measured for the PSL standards and the corresponding calculated mean scattering cross-sections. The error bars represent the histogram modes' half widths and scattering cross-section standard deviations respectively. The red brown solid line shows the best fit, the shaded area the range between the 16 and 84th percentiles of the fit function PDFs. To allow for direct comparison with the results of the new method the right hand plot shows the parameter solution ensemble in the same way as in Fig. A1.