# Peer review of "On the parametrization of optical particle counter response including instrument-induced broadening of size spectra and a self-consistent evaluation of calibration measurements"

_Atmospheric Measurement Techniques, 2017_

## Referee Comment (RC1) · D. Baumgardner (Referee) · 20 Apr 2017

From the perspective of an engineer and instrument developer, the authors have developed a very convincing processing methodology that seems to be able to account for the broadening of ambient size distributions as a result of uncertainties intrinsic to the basic operating principles of single particle light scattering instruments. The methodology that is presented offers an improvement over other methodologies that have also recognized the limitations of these type of instruments but stopped short of introducing a more complete list as in done in the current study. I am a firm believer that if we

know how to improve the accuracy and reliability of a measurement, then we should do it, regardless of how small of an incremental improvement it might be. That is the engineering purist in me that supports that view.

From the perspective of a scientist, I ask these questions of the authors,

1) The proposed methodology would seem to require quite a bit of effort and I have a very large set of measurements of atmospheric aerosols with a very wide variety of composition, shapes and sizes. How much computational time will it take for me to analyze 20,000 spectra of BC, BrC, OC, dust, bioaerosols, sea salt and inorganic compounds?

2) The focus of my research is to model the impact of these different aerosol populations on climate. After I have corrected my 20,000 spectra, I wish to put them into my climate model. To do this I need to compute extinction coefficients, single scattering albedos and asymmetry factors, all size, composition and shape dependent. Does your methodology tell me what indices of refraction and shapes were used to come up with the inverted size distribution? If not, will the uncertainty in my derived optical properties be any smaller using the corrected size distributions than if I had just used the measurements as they are and made estimates of the particle optical properties? If the answer is yes, then provide me with a quantitative measure of how much improvement.

In the introduction, the authors state: "The size distribution of aerosol particles is a key property to understanding the impact of aerosols on human health and Earth's climate.". This is the first and last time the environmental importance of aerosols is mentioned. In order to bring closure to this study and have scientific relevance, this study needs to convince the reader and potential user of the technique that there is a real need to apply this technique before using the measurements in scientific research. I understand that AMT is specifically focused on measurement techniques; however, in the journal section on Aims and Scope, this sentence makes clear that "Papers submitted to AMT must contain atmospheric measurements, laboratory measurements

relevant for atmospheric science. . ..". It is the joint responsibility of the authors, the editor and the reviewers to make sure that published papers meet these aims and scope. Without the additional answers to the questions I have posed, I am not yet convinced of this study's scientific relevance.

---

## Referee Comment (RC2) · Anonymous Referee #2 · 14 Jun 2017

In the submitted manuscript the authors introduce a new approach to model single particle light scattering instrument response by implementing a simple parametrization of the broadening effect, show a self-consistent way to evaluate calibration measurements, and outline how to obtain realistic uncertainty estimates for OPC size distributions.

Does the paper address relevant scientific questions within the scope of AMT? Yes, it does. In my opinion the scientific community have need of such an insight to the operation of this simple light scattering based measurement method, so that to better

understand and correctly interpret measured data from OPCs. Besides presenting the new model the authors introduce existing concepts for sizing and calibration evaluation and compare them using measured data for two commercial OPCs involved in the SALTRACE campaign aimed to investigate atmospheric aerosols. The presented information and methodology is especially useful for scientists performing and/or evaluating atmospheric measurements by instrumentation based on the OPC technique.

Does the paper present novel concepts, ideas, tools, or data? Yes, it does. The authors present a new methodology that is able to describe the broadening of the measured size distribution of ambient aerosols raised from the intrinsic nature of single particle light scattering instruments. The presented approach improve the accuracy of measured size distributions and gives an estimation on the uncertainties of OPC measurements.

Are substantial conclusions reached? Yes, they are. A new model has been developed, which help to improve the accuracy of size distribution measurements and help to give an estimate on the uncertainties of OPC measurements. The new method further correctly predicts the size-dependence of OPC counting efficiency. Besides presenting the new model the authors introduce existing concepts for sizing and calibration evaluation and compare them using measured data for two commercial OPCs involved in the SALTRACE campaign. The new method succeeds in modeling the measured histogram widths correctly.

Are the scientific methods and assumptions valid and clearly outlined? Yes, they are. The scientific methods and assumption are clearly described and supported by measured data as well.

Are the results sufficient to support the interpretations and conclusions? Yes, they are.

Is the description of experiments and calculations sufficiently complete and precise to allow their reproduction by fellow scientists (traceability of results)? Yes, it is.

Do the authors give proper credit to related work and clearly indicate their own new/original contribution? Yes, they do. The submitted manuscript contains 47 references from 1908 (1763) to 2016 which covers the state of the art on the presented field. The authors own results are clearly separated in the text.

Does the title clearly reflect the contents of the paper? Yes, it does.

Does the abstract provide a concise and complete summary? Yes, it does.

Is the overall presentation well structured and clear? Yes, it is.

Is the language fluent and precise? Yes, it is.

Are mathematical formulae, symbols, abbreviations, and units correctly defined and used? Yes, they are.

Should any parts of the paper (text, formulae, figures, tables) be clarified, reduced, combined, or eliminated? No, they shouldn't.

Are the number and quality of references appropriate? Yes, they are.

Is the amount and quality of supplementary material appropriate? Yes, it is.

My comments and questions to the authors are the following: The proposed method requires a lot of computations (modeling) and measurements which requires skilled persons, a well equipped laboratory and a considerable amount of working hours. As I understand, the data set obtained using this method is valid for that moment, and needs an update when the instrument response changes (e.g. degradation of laser power or contamination on the optics). Do you see a way for the automatization of the proposed methodology?

---

## Author Comment (AC1)

**Reply to the reviews of the manuscript "On the parametrization of optical particle counter response including instrument-induced broadening of size spectra and a self-consistent evaluation of calibration measurements" by A. Walser et al.**

We wish to thank the two reviewers for carefully reading our manuscript, for acknowledging the advantages introduced by the presented methods and for providing feedback which helped us to improve the manuscript. In the following, the questions and comments raised by the reviewers are marked in blue. Our answers are written in black and include a description of changes done to the manuscript in separate paragraphs.

Reviewer #1:

The proposed methodology would seem to require quite a bit of effort and I have a very large set of measurements of atmospheric aerosols with a very wide variety of composition, shapes and sizes. How much computational time will it take for me to analyze 20,000 spectra of BC, BrC, OC, dust, bioaerosols, sea salt and inorganic compounds?

For the application of the proposed size distribution (SD) retrieval method to atmospheric OPC measurements this is a valid question. The computational time required to gain a representative final SD solution ensemble via the proposed method depends on the number of Monte Carlo iterations, itself scaling with the number of input-parameter PDFs that need to be sampled. Moreover, it strongly depends on the CPU, code implementation and -parallelization etc. In consequence, it is impossible to give generally valid numbers here. For the presented SD retrieval results, computational time on a dual-core i7 CPU @ 2.9 GHz (without parallelization) was in the order of an hour, with still considerable room for improvement. Despite the enhanced computational cost, we think that extra effort should not prevent users from deriving more precise SD uncertainty estimates.

We also would like to clarify that the main focus of this manuscript is to introduce

a) a new parametrization of OPC response including a description of instrument-specific spectral broadening and
b) a more intuitive/conclusive way to evaluate OPC calibrations in the context of this new model.

As a side topic, we further outline how this framework could be used to thoroughly propagate uncertainties in all instrument- and particle-specific properties in a self-consistent way to finally yield improved SD uncertainty estimates (PDF ensembles). However, the model parameters (and their PDFs) resulting from the presented approach are not firmly connected to the

proposed retrieval framework, but can also be used to derive the SD via any other existing method.

Independently from the SD retrieval, the calibration evaluation described in this manuscript needs to be performed only once after each calibration measurement to obtain the OPC model parameter PDFs. In this case, the additional computational effort is, hence, negligible.

*Changes to manuscript*
The main purpose of this study is now clarified also in the caption of Fig. 1.

The focus of my research is to model the impact of these different aerosol populations on climate. After I have corrected my 20,000 spectra, I wish to put them into my climate model. To do this I need to compute extinction coefficients, single scattering albedos and asymmetry factors, all size, composition and shape dependent. Does your methodology tell me what indices of refraction and shapes were used to come up with the inverted size distribution? If not, will the uncertainty in my derived optical properties be any smaller using the corrected size distributions than if I had just used the measurements as they are and made estimates of the particle optical properties? If the answer is yes, then provide me with a quantitative measure of how much improvement.

With your first question you raise an important point and our answer is: Yes, it does. As the proposed SD retrieval is based on a Monte Carlo method, it is possible to write out/save all picks from the input-parameter PDFs for each Monte Carlo iteration, which later allows for a one-to-one mapping between any SD solution ensemble member and the corresponding parameter value picks (e.g. the used refractive index).

Speaking of parameters calculated from the SD (e.g. extinction coefficient, effective particle diameter etc.) reveals another major advantage of gaining a SD solution ensemble in the proposed way, i.e. it allows for simple further propagation of SD uncertainties via the solution ensemble members. This is not possible in a similar manner using other (direct) SD derivation methods.

Concerning the second question: Using the proposed retrieval method the uncertainty ranges of the SD and related quantities will not necessarily be smaller, but we claim they will be more realistic due to the thorough uncertainty propagation. We consider it essential to give reliable uncertainty estimates. If and to what extent SD solutions (in view of absolute values and uncertainties) obtained by the combination of the new OPC response model and the proposed retrieval method will differ from that of other methods depends on several factors, as for example

- the individual OPC properties (e.g. degree of spectral broadening),
- the actual SD shape (e.g. if it features narrow modes or not) and
- the previous knowledge on the aerosol properties (e.g. the uncertainty of the particles' refractive indices)

It may be true and is already stated in the manuscript (end of Sect. 4) that

a) for broad atmospheric SDs the effect of instrument-induced spectral broadening on the measured distribution's width may be small and
b) other, non-instrumental uncertainties (as the one for the refractive index) may dominate the overall SD uncertainty.

In such cases it might be lavish in view of computational effort to perform a comprehensive propagation of all parameter uncertainties (even the irrelevant ones). However,

a) the new OPC response parametrization for the first time allows to realistically model OPC response in all situations, even for SDs that are narrow with respect to the degree of spectral broadening and
b) within the proposed (Monte Carlo) SD retrieval framework the user can decide for each individual case which initial sources of uncertainty need to be sampled and which are redundant (which can be tested by sensitivity studies).

*Changes to manuscript*
Section 2.5.3 and 5:
Additional advantages of the proposed SD retrieval method, going beyond a thorough SD uncertainty assessment, are now discussed in the text, including the direct mapping between SD solutions and the corresponding parameter samples (e.g. facilitating parameter sensitivity studies) and the simplified further propagation of SD uncertainties.

Section 2.3:
The relevance of instrument-induced spectral broadening for atmospheric research is emphasized by adding examples of narrow SDs that may be significantly influenced by this effect.

In order to bring closure to this study and have scientific relevance, this study needs to convince the reader and potential user of the technique that there is a real need to apply this technique before using the measurements in scientific research. […] in the journal section on Aims and Scope, this sentence makes clear that "Papers submitted to AMT must contain atmospheric measurements, laboratory measurements relevant for atmospheric science…". It is the joint

responsibility of the authors, the editor and the reviewers to make sure that published papers meet these aims and scope.

We agree that clarifying the scientific relevance of the presented study is important and appreciate the reference to the aims and scope of AMT ("Papers submitted to AMT must contain atmospheric measurements, laboratory measurements relevant for atmospheric science, and/or theoretical calculations of measurements simulations with detailed error analysis including instrument simulations."). In accordance with the second referee we are convinced that the study is of relevance for atmospheric OPC measurements and meets the aims and scope of AMT for the following reasons:

To date, SDs derived from OPC measurements often lack a thorough uncertainty analysis. The presented OPC response model and the self-consistent calibration evaluation method partly closes this gap by offering for the first time a realistic parametrization of this response and, more importantly, realistic uncertainty estimates for the latter. The proposed SD retrieval method further outlines how to propagate these instrument-specific uncertainties and the uncertainties in all other important parameters into final SD uncertainties. Besides yielding improved SD uncertainty estimates, this method additionally permits to easily carry these uncertainties further, e.g. to obtain proper uncertainties for SD-dependent quantities like the effective particle diameter.

Apart from the study's relevance with respect to a realistic SD uncertainty analysis, instrument-specific spectral broadening may significantly bias OPC-derived SDs when disregarded. This is demonstrated particularly in Fig. 9 for the example of narrow SDs. Here, we give quantitative numbers revealing that the new OPC response model considering the instrument-induced broadening of size spectra can lead to substantial improvements. Although atmospheric SDs will usually feature broader widths than the presented example samples and will, hence, be less modified by spectral broadening, there are examples for atmospheric research where this effect is relevant (e.g. ice particle residuals in (contrail) cirrus or aerosol chamber experiments). Taking account of spectral broadening, the new method allows correctly retrieving SDs for all situations.

The application of the presented methods to atmospheric aerosol measurements will be the substance of future studies and publications.

*Changes to manuscript*
Figure 9:
The figure, showing SD retrieval results for exemplary measurements of aerosol samples with narrow size spectra, is complemented by a visualization of the SD solutions in common $dn/dD$ vs. $D$ space ($n$: particle concentration, $D$: particle diameter). This additional representation

demonstrates potential OPC measurement biases associated with (instrument-induced) spectral broadening in a more intuitive way.

Section 2.3 and 5:
The relevance of spectral broadening for atmospheric research  is now emphasized in the text.

Reviewer #2:

The proposed method requires a lot of computations (modeling) and measurements which requires skilled persons, a well-equipped laboratory and a considerable amount of working hours. As I understand, the data set obtained using this method is valid for that moment, and needs an update when the instrument response changes (e.g. degradation of laser power or contamination on the optics). Do you see a way for the automatization of the proposed methodology?

We confirm that regular OPC calibrations involve time and effort. Nevertheless, they are inevitable to ensure data quality and to avoid systematic sizing biases. Some OPCs allow the user to assess the need for recalibration by monitoring reference parameters (e.g. light source intensity), thus giving indications of instrument response changes. Tracking such reference values can help to minimize the number of necessary recalibrations. Further, routine measurements of a small subset of particle standards can be used to judge the instrument performance and decide whether an actual recalibration is required.

In contrast to the calibration measurements that will hardly be completely automatable, the evaluation via the presented method can be automated (to a large extent) when the calibration data (format) is kept consistent and the method is implemented in a suitable way.

---

## Author Response (AR2)

**Reply to the editor review of the manuscript "On the parametrization of optical particle counter response including instrument-induced broadening of size spectra and a self-consistent evaluation of calibration measurements" by A. Walser et al.**

We wish to thank the editor for carefully reading our manuscript and for providing valuable feedback which helped us to improve the manuscript. In the following, the questions and comments raised by the editor are marked in blue. Our answers are written in black and include a description of changes done to the manuscript in separate paragraphs.

Reviewer #1 raised questions about assessing the value of this new approach for practical determinations of particle size distribution in ambient conditions with unknown mixtures of aerosol species with differing index of refraction as well as shape. Clearly, without additional information about the actual particle populations, the new approach cannot realistically reduce uncertainty about the intrinsic uncertainties in the ambient particles. Your response made this clear, yet I am concerned that the reader of the revised manuscript may still suffer from some misunderstanding. Reviewer #1's question appeared to include the suggestion that your approach would provide an *optimized* return of index of refraction. This is not the case - it will only return a set of possible solutions without any "relative value" associated with each solution. Please add short discussion making this explicitly clear to the (new) reader, perhaps with some guidance about whether you recommend that this approach be performed for all data collected, for special sampling opportunities, or for "spot assessments" of uncertainties in sub-sets of larger data sets.

We never intended to cause any misunderstanding about the capabilities of the proposed size distribution retrieval method and we agree that it is important to make its limitations clear to the reader. The method allows for a thorough propagation of all initial parameter PDFs into a final size distribution PDF, but it cannot rate the value of individual parameter picks, i.e. it cannot refine the initial parameter PDFs/uncertainties.

*Changes to manuscript*
Section 2.5.3:
We complemented the discussion about the benefits of the proposed size distribution retrieval method by a sentence clarifying the above-said. We further added a recommendation for when to use the proposed approach.

The use of "spectral" (eg at lines 18 - 22 of page 6, and line 10 of page 10, etc.), could be confusing to some readers (who may think of wavelength-dependencies). Perhaps "signal" broadening? or "size broadening" would be clearer? Or simply "light intensity variations"?

We appreciate this hint and the proposed alternatives.

*Changes to manuscript*
The (misleading) term "spectral broadening" is replaced by "signal broadening".

In section 2.5.1, the discussion about light intensity variability focused on in Section 2.3 is rehashed; later you mention (page 9) changes in scattering geometry again. Please consider if these "partial" references should simply be redirects to a complete discussion of all the sources in Sec 2.3. Perhaps this section should also include a wider range of possible broadening source. Ones that occur to me include: the time response of the detectors/sensing system to particles passing through the lit region at different speeds, the temperature dependent variation in detector sensitivity, and the altitude dependent variations in background-noise that are possible with some systems…others were referenced at other points in the paper.

We concur that the rehashed discussion about light intensity variability in Sect. 2.5.1 is redundant. Yet, the subject of the discussion on page 9 isn't instantaneous broadening of size spectra ("signal broadening") but tries to give possible explanations for remaining (temporal) deviations between measured and modeled OPC response, e.g. induced by light source intensity fluctuations (on larger time scales). This discussion is not really belonging to the signal broadening topic treated in Sect. 2.3. Therefore, we think it is better kept separated from the latter.

*Changes to manuscript*
Section 2.3 and 2.5.1:
The detailed discussion about the non-uniformity of light intensity in the OPC sampling volume and its implications for the width of recorded size spectra is now included in Sect. 2.3. The respective part in Sect. 2.5.1 is replaced by a reference.

Section 2.5.2 (Page 9):
The discussion of potential reasons for remaining (time-dependent) model deviations is kept in place, but the differences to the signal broadening topic (Sect. 2.3) are now made explicitly clear. Further, some of the proposed additional sources for time-dependent OPC response fluctuations are included.

[revised manuscript text omitted]